

# On the relation between fractional charge and statistics

**Thors Hans Hansson**[1⋆], **Rodrigo Arouca**[2†] **and Thomas Klein Kvorning**[3‡]

**1** Fysikum, Stockholm University, Stockholm, Sweden
**2** Department of Physics and Astronomy, Uppsala University, Sweden
**3** Royal Institute of Technology, Stockholm, Sweden

⋆ hansson@fysik.su.se , † rodrigo.arouca@physics.uu.se , ‡ kvorning@kth.se

## Abstract

We revisit an argument, originally given by Kivelson and Roček, for why the existence of fractional charge necessarily implies fractional statistics. In doing so, we resolve a contradiction in the original argument, and in the case of a $\nu = 1/m$ Laughlin holes, we also show that the standard relation between fractional charge and statistics is necessary by an argument based on a t'Hooft anomaly in a one-form global $\mathcal{Z}_m$ symmetry.

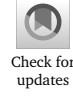

In the early days of quantum Hall physics, the connection between quantized Hall conductance, fractional charge, and fractional statistics was by no means obvious. One of the landmarks in the understanding of the fractional quantum Hall effect was Laughlin's 1983 argument [1] for why a quantized quantum Hall conductance implies fractional charge. A likewise early, but less known, argument for why fractional charge implies fractional statistics was given in 1985 by Kivelson and Roček [2]. Importantly, their argument applies more generally than to just quantum Hall systems, since it relies on only minimal theoretical assumptions. These arguments marked the beginning of the exploration of topological quantum matter.

The contemporary understanding of the topological properties of quantum Hall fluids is in terms of topological quantum field theories, typically involving Chern-Simons terms [3] and closely connected conformal field theories [4]. There is also a well-established connection to the mathematics of braided tensor categories; for a pedagogical introduction, see Ref. [5]. The extensive work on various concrete wave functions [1, 6, 7] also fits nicely into these general theoretical frameworks, while simultaneously providing essential information about the excitations and edge dynamics of many observed quantum Hall states.

However, being aware of the general arguments based on minimal theoretical assumptions remains valuable. In this note, we revisit the Kivelson-Roček argument, find a loophole, and present a reformulation that avoids it. Interestingly, to formalize our version of the argument, we are led to invoke a 't Hooft anomaly in a global one-form symmetry.

The original argument by Kivelson and Roček was later developed by Karlhede, Kivelson and Sondhi [8], and we now recapitulate their version. Assume that we have two quasiparticles with fractional charge $\nu e$, which we shall refer to as anyons, and imagine threading a thin unit flux tube through one of them. Next, move the other particle adiabatically around the flux-charge composite. This process will result in an Aharonov-Bohm (AB) phase $2\pi\nu$. But since

the system is made entirely out of charge $e$ electrons, a unit flux is invisible; the Byers-Yang theorem [9]. Thus, there must be a compensating $-2\pi\nu$ (mod $2\pi$) braiding phase, and thus a $-\nu\pi$ exchange phase. This is the statistical phase. Note that this argument does not evoke the physics of a quantum Hall liquid. Upon closer examination, one realizes that this argument cannot be correct. Suppose that we insert $n$ fluxes instead of just one, then we would conclude that the statistical phase is $-n\nu\pi$ not $-\nu\pi$.

That topological interactions and fractional charge go hand in hand can already be seen from a simplified version of the Kivelson-Roček argument. Assume that we have a low-energy eigenstate of a (bulk) gapped Hamiltonian which hosts a particle, with charge $\nu e$, and suppose that we adiabatically thread a flux through some point far away from this excitation. Because of the gap, local observables remain unchanged except in a region close to where the flux is threaded. However, the eigenvalue of a *non-local* gauge invariant operator $U$ that implement a process where the particle encircles the flux, will change continuously from unity to $e^{2\pi i\nu}$ as the flux increases from zero to a flux quantum. However, a unit flux must be invisible. But for this to be true, $U$ must have the same eigenvalue, so we conclude that an excitations is created at the point of the flux insertion which has a braiding phase of $e^{2\pi i\nu}$ relative to the original fractionally charged particle. Note that this argument tells us nothing about the mutual statistics between two of the original particles, but only that the state supports excitations with non-trivial braiding relative to the fractionally charged particles. In principle, these could have trivial statistics, if the state supports additional uncharged anyons alongside the fractionally charged particles. Then their fusion product—hypothetically the lowest-energy excitation—could be fractionally charged particles with trivial braiding. However, if we assume that the fractionally charged particles are the *only* topologically nontrivial particles, then their statistics will be fixed by a modified Kivelson-Roček argument, as we now show.

First, we recall a version of Laughlin's argument for fractional charge. If we adiabatically thread a thin flux through a $\sigma_H = \nu\sigma_H^0$ FQH state, the induced electromotive force will push the liquid radially outwards. For $n$ unit fluxes, $n\phi_0$, the Hamiltonian is back to its original form up to a regular gauge transformation. This means that the flux can be removed, and we are left with a localized fractional charge; the compensating fraction being pushed to the edge of the sample. Next, repeat the procedure to insert another charge flux composite. We now show that there is an obstruction against removing this second flux. For this, note that the world line of the charge-flux composite can be thought of as a t' Hooft and a Wilson loop superimposed on each other, $WB_\nu(C) \equiv W_\nu(C)B(C)$; such superpositions have been considered earlier by Itzhaki [10]. Here, $W(C) = \exp[i\oint_C dX^i A_i(x)]$, and the t'Hooft loop, $B(C)$, implements a gauge transformation that has a $2\pi$ phase jump at an area spanned by the curve $C$.

Now consider a correlation function $\langle WB_\nu(C_1)W_\nu(C_2)\rangle = e^{2\pi\nu L(C_1,C_2)i}$, where $L(C_1,C_2)$ is the linking number of the two loops [11]. This phase is just the AB phase picked up when the $\nu e$ charge encircles the unit flux vortex. It is now clear that we cannot just remove the flux since that would change the value of a gauge invariant correlation function. On the other hand, removing a unit flux in a system consisting entirely of charge $e$ electrons is a regular (global) gauge transformation. Thus, to remove the second flux to get a description entirely in terms of fractionally charged quasiparticles, we must add a compensating statistical phase to restore the original value of the correlation function. Note that the sign of this phase is opposite to the one in the original Kivelson-Roček argument. This obstruction to a regular global gauge transformation is reminiscent of a global anomaly. In this case, however, it is not the partition function that acquires an anomalous phase, but rather certain gauge invariant correlation functions related to braided loops. The connection to anomalies is, however, somewhat subtle and will be discussed below. An argument very similar in spirit to the above was given earlier by Feldman and Halperin [12].



It is now clear what happens if we insert $n$ fluxes instead of only one. This process will generate a charge $n\nu e$, and the mutual statistics phase with respect to the original $\nu e$ quasiparticle will be $n\nu\pi$ as required by consistency.

To give a formal version of the above heuristic argument, we first show how to form flux-charge composites using a topological field theory construction. We then show that this theory is consistent (or more precisely, anomaly free) only for certain combinations of fluxes and charges, which explains why "removing fluxes", in general, is not allowed. To achieve this, we consider two kinds of charges, which we refer to as charge and vortex, respectively, with the corresponding currents $j$ and $j_v$, where the former couples to the electromagnetic field $A$. The AB phase related to moving a charge $q$ around an $n$-vortex is encoded in the following BF Lagrangian,

$$\mathcal{L}_{AB} = \frac{1}{\pi}adb - qaj - nbj_v - eqAj, \tag{1}$$

where we also added the coupling to the external field. Next, consider a composite of a charge $q$ and $n$ vortices, i.e. put $j_v = j$; we shall comment on this identification later. Introducing the new variables

$$\tilde{a} = nb + qa, \tag{2}$$
$$\tilde{b} = nb - qa,$$

we can rewrite (1) as

$$\mathcal{L}_{AB} = \frac{1}{4\pi qn}\tilde{a}d\tilde{a} - \tilde{a}j - eqAj, \tag{3}$$

where we neglected a boundary term and also the term $\sim \tilde{b}d\tilde{b}$ since $\tilde{b}$ does not couple to the current. Specializing to a Laughlin hole obtained by inserting one flux quantum we have $q = 1/m$ and $n = 1$, gives

$$\mathcal{L}_{1/m} = \frac{m}{4\pi}\tilde{a}d\tilde{a} - \tilde{a}j - \frac{e}{m}Aj, \tag{4}$$

which is the hydrodynamic action for a $\nu = 1/m$ Laughlin state, known to give a statistical interaction to the sources with a statistical exchange parameter $\theta = \nu\pi$ [13]. Thus, we have shown that the fundamental, i.e. the charge $e/m$, Laughlin holes created by insertion of a unit flux can alternatively be described as fractionally charged point-like anyons where the fluxes are removed.

At first sight, this result looks strange. We started from two composite particles, each with a unit flux and a charge $q = \nu e$, and thus we would expect a braiding phase $\gamma = 2 \times 2\pi\nu = 4\pi\nu$, while (4) gives $\gamma = 2\pi\nu$, which is the correct result; fermions have $\nu = 1$, and gives a unit braiding phase i.e. a sign under exchange. This apparent contradiction is resolved by noticing that the identification $j_v = j$ changes the configuration space, and such identifications will indeed change statistics. The most famous example is given in the original article on fractional statistics by Leinaas and Myrheim [14], and for the present case, the explanation is given in Ref. [15]. Related arguments for the absence of the factor of 2 in a somewhat different context is given in Ref. [16].

Note that in the heuristic argument, which does not involve any identification of currents, we cannot start from an assembly of charges threaded by fluxes, but must proceed stepwise: Starting from the already explained two-body case, we can move to three particles by introducing a new flux-charge composite in the presence of the two $e/m$ charges obeying $\pi/m$ fractional statistics. We can then again remove the flux at the expense of introducing fractional statistics between the new charge and the two already present. This way of reasoning can be extended to any number of particles, but it is not clear how such a recursive argument could be formulated rigorously. To do so, we return to the formal argument based on Eqs. (1–4).

For $n \neq 1$, we first note that taking $n = m$ gives a trivial braiding phase as expected, while $1 < n < m$ amounts to having a non-integer level number, signaling that the theory is not consistent. In fact, it gives a braiding phase $\gamma = 2\pi\nu n$, which depends on $n$ contrary to the Byers-Yang theorem. However, if we insert $n$ fluxes in a $\nu = 1/m$ Laughlin liquid we get the charge $q = n/m$, so after the rescaling $\tilde{a} \to n\tilde{a}$ (3) becomes,

$$\mathcal{L}_{1/m} = \frac{m}{4\pi}\tilde{a}d\tilde{a} - n\tilde{a}j - \frac{ne}{m}Aj\,, \tag{5}$$

which is consistent, and give a braiding phase $\gamma = 2\pi\nu n^2$ as expected for a composite of $n$ anyons with $q^\star = e/m$. Note that in this case, different $n$ describes different particles, and it should be clear that these results again are in full agreement with our initial heuristic arguments.

We now give a formal argument for why it is inconsistent to remove the flux from a composite of one charge and several fluxes, and we now show that such a theory is anomalous. For this, we first notice that taking $n > 1$ in (3) we have a $U(1)$ CS theory with a fractional level number, and when defined on a closed manifold, such a theory has a global $U(1)$ anomaly. (For a simple explanation, see Ref. [17].) This does, however, not fully reflect the heuristic arguments given at the beginning of this article, where there was no need to assume a closed manifold. We now give an alternative way to understand the anomaly that works also for an open manifold. To do so, return to (3) and set the charge $q = 1/m$,

$$\mathcal{L}_{1/m} = \frac{m}{4\pi n}\tilde{a}d\tilde{a} - \tilde{a}j - \frac{e}{m}Aj\,. \tag{6}$$

Following Ref. [18], we note that this theory has a global one-form $\mathscr{Z}_m$ symmetry that is generated by the Wilson lines,

$$U_{e^{2\pi ik/m}}(M^{(1)}) = \exp\left(ik \oint_{M^{(1)}} \tilde{a}\right)\,, \tag{7}$$

where $M^{(1)}$ is an one-manifold, and $k = 1\ldots m-1$. The Wilson loops corresponding to the source $j$ in (6) are themselves generators of this algebra, and as stressed in Ref. [18], they are also charged under the symmetry $U_{e^{2\pi ik/m}}(M^{(1)})$ since from (6) it follows that

$$U_{e^{2\pi ik/m}}(M^{(1)}) \exp\left(i \oint_{\mathcal{C}^{(1)}} \tilde{a}\right) = e^{\frac{2\pi ikn}{m}} \exp\left(i \oint_{\mathcal{C}^{(1)}} \tilde{a}\right)\,, \tag{8}$$

where $M^{(1)}$ is a small loop around $\mathcal{C}^{(1)}$. For $n = 1$, $U_{e^{2\pi ik/m}}(M^{(1)})$ generates the $\mathscr{Z}_m$ symmetry corresponding to the $m$ ground states on a torus. In the case of $n$ fluxes with $n > 1$, we now ask whether or not they can be removed with impunity. If they can, we should get the same result for any correlation function of the charged Wilson loops if we insert any number of $U_{e^{2\pi ik/m}}(M^{(1)})$ and then average over $n$ for fixed $k$, which amounts to gauging the symmetry. However, from (8), we see that such an average will give zero, which is a consequence of the symmetry having a t' Hooft anomaly as explained in Ref. [18]. From this, we conclude that for flux charge composites with more than one flux per charge, it is not possible to remove the flux, which was what we set out to prove.

The essence of the above rather formal argument can be understood in simpler terms that connects directly to our first heuristic argument: The introduction of the loop $\mathcal{C}_1$ changes the topology of space-time; think of it as a thin closed tube. The presence of such a tube allows us to introduce fluxes just as one does in the holes of a torus when one proves that the level number of the CS theory must be integer.

Concerning the internal structure of the quasiparticles: If we take the composite picture at face value and assume that the flux is separated from the charge, the situation is well understood [19, 20]; rotating the system an angle $2\pi$ amounts to braiding the flux and the charge *within* the composite. The corresponding phase angle $\pi/m = \pi\nu$, is precisely what is expected for particle with spin $s = \nu/2$, which is the generalized spin-statistics relation [21]. Also, in the context of CS theory, the fractional spin can be calculated as the orbital angular momentum as first shown in Ref. [22]. It is not clear, however, how these arguments can be formalized to extend the description based on (1), although a natural starting point would be to formulate that theory on a spatially curved surface.

Finally, we stress that our main result is that fractional charge necessarily implies fractional statistics as originally claimed in Refs. [2, 8]. This result is independent of assumptions about the internal state of flux-charge composites, which, however, is important when discussing the spin. In particular, we showed that starting from the BF lagrangian (1), which was written just to describe the AB phases related to charges encircling fluxes, we can, by identifying the charge and vortex currents, derive the hydrodynamic theory (3), and (5) which does not involve any fluxes, and correctly incorporates the fractional statistics between point-like anyons.

## Acknowledgments

We thank Bert Halperin, Jainendra Jain and Steve Kivelson for discussions, and comments. RA thanks the KOMKO group of Fysikum for the hospitality.

**Funding information**   RA acknowledges financial support from the Knut and Alice Wallenberg Foundation through the Wallenberg Academy Fellows program KAW 2019.0309 and project grant KAW 2019.0068. T. K. K. acknowledges funding from the Wenner-Gren Foundations.

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
