# Peer review of "On the relation between fractional charge and statistics"

_SciPost Physics, doi:SciPost Phys. 18, 197 (2025)_

## Round 1 · Referee Report · Anonymous (Referee 1) · 2025-2-17

Report

This is an excellent paper. The authors revisit an old argument which argued that fractional charge implies fractional statistics and showed that had a serious loophole. They they furnished a complete proof by using both heuristic arguments and modern methods of generalized symmetries and quantum anomalies . The paper is well written and conceptually self-contained. It clarifies the physical linkage between these concepts . It is clearly a very valuable new addition to the literature on this subject.

Recommendation

Publish (surpasses expectations and criteria for this Journal; among top 10%)

  • validity: top
  • significance: top
  • originality: top
  • clarity: top
  • formatting: perfect
  • grammar: perfect

Author:  Rodrigo Arouca  on 2025-04-01  [id 5325]

(in reply to Report 1 on 2025-02-17)

Thank you for your kind and thorough review. We truly appreciate your positive feedback.

---

## Round 1 · Referee Report · Anonymous (Referee 2) · 2025-2-21

Strengths

1- closes the loophole in the original argument on the relation between fractional charge and fractional statistics in FQHE and similar phases of matter.

2- establishes relation between the problem and t'Hooft anomalies for 1-form symmetries

3- clearly written

4- gives necessary references

Weaknesses

1- Possible generalizations of the presented argument are not discussed (see the report)

Report

The manuscript revisits old arguments relating fractional charge and statistics in gapped systems such as FQHE. The authors show how the loophole in the classical argument can be closed. The key ideas are: - assume that the system does not support neutral excitations with fractional statistics.
- Show that there is an obstruction in removing integer fluxes from composite objects with several fluxes. - The latter is shown by showing the inconsistency of the obtained theory (after removal) due to t'Hooft anomaly for the global Z_m symmetry present in the Chern-Simons action for fractional charge of elementary excitations.

I find the arguments convincing and worth publishing in SciPost Physics. There are few open questions that are only partially addressed/mentioned in the manuscript. - The internal structure of composite objects and fractional spin. It seems plausible that coupling additionally the system with gravitational background one could generalize the argument proving the fractional charge of the fractional excitations. It probably can be done using constructions discussed in X. Wen and A. Zee, Phys. Rev. Lett., 69, 953 (1992). and the ideas of framing and framing anomalies G. Y. Cho, etal. Phys. Rev. B, 90, 115139 (2014), A. Gromov etal, Phys. Rev. Lett. 114, 016805 (2015). - Let us assume that there are multiple species of fractionally charged particles and particles with fractional statistics (including possibly neutral particles). It seems that the t'Hooft anomalies discussed in the manuscript would provide some constraints on possible content in terms of charges, spins and statistics of excitations. It might be interesting to see these more general relations.

While manuscript would benefit, I think, from discussions of these generalizations, it might be a subject of a separate work.

Requested changes

1- The paragraph starting with "Concerning the internal structure of the quasiparticles..." on page 5 is repeated twice. Please, remove one of those.

2- Few comments related to the generalizations mentioned in the report section would be appreciated but are Optional.

Recommendation

Publish (easily meets expectations and criteria for this Journal; among top 50%)

  • validity: high
  • significance: high
  • originality: high
  • clarity: high
  • formatting: excellent
  • grammar: excellent

Author:  Rodrigo Arouca  on 2025-04-01  [id 5326]

(in reply to Report 2 on 2025-02-21)

We thank you for your thorough reading of our manuscript and are pleased you find it suitable for publication in SciPost. We also appreciate you identifying the typo (which is now corrected) and your insightful suggestion to consider relations between fractional spin, statistics, and charge, rather than only the latter two. While this direction is indeed intriguing, we believe a more detailed discussion would deviate from our main message and is better suited for a separate publication.

---

## Round 2 · Referee Report · Anonymous (Referee 2) · 2025-4-25

Strengths
1- The important problem of the relationship between fractional charge and statistics is revisited with new insights arising from the ’t Hooft anomaly of a 1-form symmetry.
2- Previous heuristic arguments are discussed, and the loopholes in those arguments are carefully examined.
3- A rigorous argument is presented to explain the impossibility of having strictly local excitations with fractional charge.
2- Previous heuristic arguments are discussed, and the loopholes in those arguments are carefully examined.
3- A rigorous argument is presented to explain the impossibility of having strictly local excitations with fractional charge.
Weaknesses
1- I find the article somewhat convoluted and not sufficiently clear in its presentation.
Report
To elaborate on point 1: it would be helpful for the formal part of the manuscript to be self-contained. The argument begins with Eq. (1), after which the relevant assumptions should be clearly stated, and the “flux removal” procedure explained at the formal level. It should then be demonstrated that this removal renders the theory inconsistent due to the anomaly. While all of these elements are present in the derivation, their presentation is far from straightforward.
To conclude, I believe the manuscript is suitable for publication in SciPost Physics in its current form. However, I (and, hopefully, the readers) would appreciate a clearer exposition of the formal argument connecting fractional charge to fractional statistics.
To conclude, I believe the manuscript is suitable for publication in SciPost Physics in its current form. However, I (and, hopefully, the readers) would appreciate a clearer exposition of the formal argument connecting fractional charge to fractional statistics.
Recommendation
Publish (easily meets expectations and criteria for this Journal; among top 50%)

Author: Rodrigo Arouca on 2025-05-12 [id 5470]
(in reply to Report 1 on 2025-04-25)We thank the referee for thoroughly reading our manuscript, and making a relevant point about the presentation. We are pleased that they find it suitable for publication in SciPost. We agree that we should have explained the intent and logic of the formal derivation better, and also made a clearer separation from the initial intuitive discussion. We have reformulated the text above eq. 1 to smooth the transition between the two parts of the paper.

---

## Round 2 · List of Changes

We removed the repeated sentences indicated by one of the referees.

---

## Round 3 · Author Response

Dear Editor and Referees,
Thank you for taking the time to carefully review our manuscript and for providing thoughtful feedback. We appreciate that all referees have a positive assessment
of our work and for recommending it for publication in SciPost Physics.
Following the suggestions from Referee 3, we revised the text before the formalization of our argument, providing a clearer context. We believe this modification contributed to make our manuscript better, and we thank the referee for this suggestion.
With the above modification, we expect that our text will be acceptable for publication.
Thank you again for your time and valuable input.
Sincerely,
Hans, Rodrigo, and Thomas
Thank you for taking the time to carefully review our manuscript and for providing thoughtful feedback. We appreciate that all referees have a positive assessment
of our work and for recommending it for publication in SciPost Physics.
Following the suggestions from Referee 3, we revised the text before the formalization of our argument, providing a clearer context. We believe this modification contributed to make our manuscript better, and we thank the referee for this suggestion.
With the above modification, we expect that our text will be acceptable for publication.
Thank you again for your time and valuable input.
Sincerely,
Hans, Rodrigo, and Thomas

---

## Round 3 · List of Changes

We reformulated the text before Eq. (1) to provide a better bridge between the intuitive arguments of the beginning of the text and the formal arguments that follow later.

---

## Editorial Decision

published